# Proteomic analysis of *Sarcoptes scabiei* reveals that proteins differentially expressed between eggs and female adult stages are involved predominantly in genetic information processing, metabolism and/or host-parasite interactions

**Tao Wang**[1]*, **Robin B. Gasser**[1], **Pasi K. Korhonen**[1], **Neil D. Young**[1], **Ching-Seng Ang**[2], **Nicholas A. Williamson**[2], **Guangxu Ma**[1,3], **Gangi R. Samarawickrama**[4,5], **Deepani D. Fernando**[4], **Katja Fischer**[4]

1 Faculty of Veterinary and Agricultural Sciences, The University of Melbourne, Parkville, Australia, 2 Bio21 Mass Spectrometry and Proteomics Facility, The University of Melbourne, Parkville, Australia, 3 College of Animal Sciences, Zhejiang Provincial Key Laboratory of Preventive Veterinary Medicine, Zhejiang University, Hangzhou, China, 4 Infection and Inflammation Program, QIMR Berghofer Medical Research Institute, Brisbane, Australia, 5 School of Veterinary Science, University of Queensland, Gatton, Australia

☯ These authors contributed equally to this work.
* tao.wang1@unimelb.edu.au

## Abstract

Presently, there is a dearth of proteomic data for parasitic mites and their relationship with the host animals. Here, using a high throughput LC-MS/MS-based approach, we undertook the first comprehensive, large-scale proteomic investigation of egg and adult female stages of the scabies mite, *Sarcoptes scabiei*–one of the most important parasitic mites of humans and other animals worldwide. In total, 1,761 *S. scabiei* proteins were identified and quantified with high confidence. Bioinformatic analyses revealed differentially expressed proteins to be involved predominantly in biological pathways or processes including genetic information processing, energy (oxidative phosphorylation), nucleotide, amino acid, carbohydrate and/or lipid metabolism, and some adaptive processes. Selected, constitutively and highly expressed proteins, such as peptidases, scabies mite inactivated protease paralogues (SMIPPs) and muscle proteins (myosin and troponin), are proposed to be involved in key biological processes within *S. scabiei*, host-parasite interactions and/or the pathogenesis of scabies. These proteomic data will enable future molecular, biochemical and physiological investigations of early developmental stages of *S. scabiei* and the discovery of novel interventions, targeting the egg stage, given its non-susceptibility to acaricides currently approved for the treatment of scabies in humans.

**Data Availability Statement:** The data that support the findings of this study are publicly available from PRIDE data repository (https://www.ebi.ac.uk/pride/) with the identifier (PXD032148).

**Funding:** Support from the National Health and Medical Research Council (NHMRC) of Australia (K.F., R.B.G. and D.D.F.) and the Australian Research Council (ARC; R.B.G., N.D.Y. and P.K.K.) is gratefully acknowledged. K.F. held an NHMRC Senior Research Fellowship. The funders had no role in study design, data collection and analysis, decision to publish, or preparation of the manuscript.

**Competing interests:** The authors have declared that no competing interests exist.

## Author summary

Scabies is a neglected tropical disease caused by the parasitic mite *Sarcoptes scabiei*. The treatment and control of scabies are challenging, as there is no vaccine and the two mostly used broad-spectrum acaricides (i.e. ivermectin and permethrin) do not kill the key developmental stage (egg) of the mite that enables the re-establishment of infection. The availability of a well-assembled genome for *S. scabiei* now provides a foundation to explore the molecular biology, biochemistry and physiology of this mite. Here, we characterised the first somatic proteome of key developmental stages of *S. scabiei* using high throughput LC-MS/MS. Bioinformatic analyses of proteomic data indicate that proteins expressed differentially between egg and female adult stages are mainly involved in biological pathways or processes, such as genetic information processing, energy (oxidative phosphorylation), nucleotide, amino acid, carbohydrate and/or lipid metabolism in the mite. These proteomic data should underpin further investigations of early developmental stages of *S. scabiei* with a focus on identifying novel intervention targets for scabies.

## Introduction

Scabies is a disease caused by the parasitic mite *Sarcoptes scabiei*, and is one the commonest dermatological disorders worldwide, causing major morbidity, disability, stigma and poverty in people [1]. Of the 15 most burdensome dermatologic conditions, evaluated in disability-adjusted life years [DALYs], scabies ranks higher than keratinocyte carcinoma and melanoma [2]. Importantly, scabies can be a major contributing factor to life-threatening *Staphylococcus aureus*-bacteraemia and severe post-streptococcal sequelae [3–5], such as glomerulonephritis, rheumatic fever and/or heart disease (RF/RHD), exacerbating the scabies burden [6].

There is no vaccine against scabies, and only few broad-spectrum acaricides, including ivermectin and permethrin, are approved for use in humans [1]. Available drugs kill the motile parasite stages (larvae, nymphs and adults) by interfering with the mite's nervous system and muscle function. However, control is challenging because the eggs of *Sarcoptes* are not susceptible to treatment, and the drugs have short half-lives in skin tissues. This means that eggs can survive treatment, and readily hatch and continue to perpetuate infection and the life cycle. Resistance in *Sarcoptes* against acaricides is emerging [7], emphasising the urgency to find novel and effective acaricides for the treatment and clinical management of scabies, built on a solid understanding of these mites, their relationship with animal host(s) and parasitism at the molecular and biochemical levels.

Advanced nucleic acid sequencing and bioinformatic technologies have enabled an unprecedented number of arthropod genomes to be decoded (e.g., i5K Consortium; http://i5k.github.io/; cf. [8]). Although draft genomes provide investigators with resources to explore parasitic arthropods at the molecular level, the expression profiles and functions of the vast majority of proteins are unknown. Emerging transcriptomic and proteomic data sets are now aiding investigations into the expression, localisation and function of genes.

While transcriptomics can quantify RNAs, such as small and messenger RNAs, advanced proteomics provides a means of identifying and quantifying proteins in whole arthropods, particular developmental stages and/or their tissues [9]. Digital resources to investigate and mine genomic and transcriptomic data sets of arthropods are now readily available. The i5K database [8] and the 'assembled searchable giant arthropod read database' (ASGARD; ref. [10]) contain relatively extensive genomic and/or transcriptomic datasets for parasitic arthropods, including mites, and FlyBase 2.0 [11] provides a wealth of information and features for

*Drosophila melanogaster* (vinegar or fruit fly)–the most-studied arthropod. However, a critical appraisal of the current literature reveals that, with the exception of selected mite species, such as the house dust mites *Dermatophagoides farinae* and *Dermatophagoides pteronyssinus* [9,12], proteomic data sets are scant for most socio-economically important parasitic mites, in spite of the major technological advances made recently in the proteomics field [13,14].

The aim of the present study was to define the proteome of key developmental stages of *S. scabiei*, using high throughput liquid chromatography–mass spectrometry (LC-MS/MS), and explore biological/cellular processes and pathways employing current genomic resources using advanced informatics [15,16]. Given the scant proteomic data available for *S. scabiei* (i.e., excretory/secretory proteins in mite faeces and pooled developmental stages–larvae, nymphs and adults; cf. [16]), defining the proteome of the key developmental stages of *S. scabiei* (particularly eggs and reproductively active females) should enable insights into this mite's unique biology and its ability to survive and maintain a complex relationship with its host animal, and could provide an avenue to discovering new targets for interventions in this and related mites.

## Methods

### Ethics statement

Animal ethics approval was granted by the QIMR Berghofer Medical Research Institute (permit nos. P630 and P2159) and the Ethics Committee of the Queensland Animal Science Precinct (permit SA 2015/03/504).

### Procurement of different stages of *S. scabiei*

*Sarcoptes scabiei* was produced on pigs (3 months of age), isolated and stored using well-established protocols [17]. Adult female (Af) mites (5 replicates of 500 individuals) were isolated from skin crusts from *S. scabiei*-infected pigs, incubated in olive oil (Remano, Aldi) at 23°C for 2 h, washed extensively in physiological saline (pH 7.4) and used immediately for protein extraction. In addition, eggs were cumulatively collected from skin crusts taken from pigs on different days. Eggs at an early stage (designated Ee) and eggs at a late stage (El) of embryonation/development (cf. [18]) were collected separately (5 replicates for each Ee ($n$ = 2,500) and El ($n$ = 1,500), washed extensively in saline and snap frozen at -80°C until use. Crude protein extracts were prepared from each of the replicate samples, freeze-dried and resuspended in 200 μl 8 M urea in 100 mM triethylammonium bicarbonate (pH 8.5) with protease inhibitor cocktail set I (Merck, Denmark).

### Extraction of proteins

Proteins were extracted from each of the five replicates for each developmental stage (i.e., Ee, El and Af). In brief, 500 μL of lysis buffer (8 M urea in 100 mM triethyl ammonium bicarbonate, pH 8.5) was added to individual samples ($n$ = 15), subjected to three freeze (-196°C)–thaw (37°C) cycles [19] and centrifuged at 10000 ×g for 30 sec, ultrasonicated (20 kHz) using a BioRuptor (10 cycles: 30 sec on and 30 sec off) in tubes on ice. Each sample was supplemented with the protease inhibitor cocktail set I (Merck, Denmark) and incubated at 23°C for 30 min. Then, samples were centrifuged at 12000 ×g for 30 min, and the supernatants collected for analyses. Protein concentrations were measured using a BCA protein assay kit (Thermo Fisher Scientific, USA).

### Digestion of proteins and LC-MS/MS analysis

In-solution digestion was carried out using an established protocol [20]. In brief, samples containing proteins (50 μg) from either the Ee, El or Af stage were reduced with 10 mM Tris

(2-carboxyethyl) phosphine (TCEP) at 55˚C for 45 min, alkylated with 55 mM iodoacetamide in the dark at 22˚C for 30 min, and double-digested with Lys-C/trypsin mix (Promega, USA) at 37˚C for 16 h (4 h for Lys-C and 12 h for trypsin digestion). The tryptic samples were acidified with 1.0% (v/v) formic acid, purified using Oasis HLB cartridges (Waters, USA). Then, samples were freeze-dried prior to re-suspension in aqueous 2% w/v acetonitrile and 0.05% w/v trifluoroacetic acid (TFA) prior to LC-MS/MS analysis.

Tryptic peptides were analysed using the Exploris 480 Orbitrap mass spectrometer (Thermo Fisher, USA). The LC system was equipped with an Acclaim Pepmap nano-trap column (Dinoex-C18, 100 Å, 75 μm x 2 cm) and an Acclaim Pepmap RSLC analytical column (Dinoex-C18, 100 Å, 75 μm x 50 cm). The tryptic peptides were injected into the enrichment-column at an isocratic flow of 5 μL/min of 2% v/v $CH_3CN$ containing 0.05% v/v TFA for 6 min, applied before the enrichment column was switched in-line with the analytical column. Solvent A was (v/v) 0.1% formic acid, 95% $H_2O$, 5% dimethyl sulfoxide and Solvent B was (v/v) 0.1% formic acid, 95% acetonitrile, 5% dimethyl sulfoxide. The gradient was at 300 nl/min from (i) 0–6 min at 3% B; (ii) 6–95 min, 3–20% B; (iii) 95–105 min, 20–40% B; (iv) 105–110 min, 40–80% B; (v) 110–115 min, 80–80% B; (vi) 115–117 min 85–3% and equilibrated at 3% B for 10 min before injecting the next sample. The Exploris 480 Orbitrap mass spectrometer was operated in the data-dependent mode, whereby full MS1 spectra were acquired in a positive mode (spray voltage of 1.9 kV, source temperature of 275˚C), 120000 resolution, AGC target of $3e^6$ and maximum IT time of 25 ms. The "top 3 second" acquisition method was used and peptide ions with charge states of 2–6 and intensity thresholds of $\geq 5e^3$ were isolated for MS/MS. The isolation window was set at 1.2 m/z, and precursors were fragmented using higher energy C-trap dissociation (HCD) at a normalised collision energy of 30, a resolution of 15000, an AGC target of $7.5e^5$ and an automated IT time selected. Dynamic exclusion was set at 30 sec.

## Protein identification and quantification

The proteome predicted for *S. scabiei* (9,174 protein entries) was previously annotated [16] using NCBI non-redundant (nr) protein database [21]. Raw mass spectrometry data were processed using MaxQuant [22] to identify and quantify peptides of *S. scabiei*. Search parameters were: a precursor tolerance of 20 ppm, MS/MS tolerance of 0.05 Da, fixed modifications of carbamidomethylation of cysteine (+57 Da) and methionine oxidation (+16 Da). The match between-run feature was activated. Proteins and peptides were accepted based on a false discovery rate (FDR) of $< 0.01$ at both the peptide and protein levels. Proteins were quantified using the LFQ value from MaxQuant employing default settings. For stage-specific identification and relative quantification comparisons, only proteins containing $\geq 2$ peptides and detected in $\geq 3$ biological replicates of at least one developmental stage were accepted. The data were normalised based on the median protein intensity at each condition. Raw data is available via the PRIDE data repository (https://www.ebi.ac.uk/pride/; accession number: PXD032148).

## Bioinformatic analyses of data sets

The UniProt repository was used for protein annotation (cellular compartment, subcellular location, transmembrane region and/or molecular function). Molecular functions of proteins were assigned according to Gene Ontology (GO) using the program InterProScan [23]. Venn diagrams were drawn using the VennDiagram package in R. Principal component analysis (PCA) and hierarchical cluster analysis (HCA) were conducted using Perseus software (v.1.6.1.1) employing default settings [24]. Sequence homology searches were conducted using BLASTP (https://blast.ncbi.nlm.nih.gov/Blast.cgi?PAGE=Proteins). Volcano plot analysis was

employed to assess differential protein expression using Perseus software (v.1.6.1.1), with the false discovery rate (FDR) and fold change (FC) set at $\leq 0.01$ and $> 2$, respectively. Biological functions were assigned to differentially expressed proteins using the Kyoto Encyclopedia of Genes and Genomes (KEGG) database [25]. KEGG pathway annotation was conducted employing KEGG BLASTP hits (E-value: $< 10–5$) and corresponding KEGG Orthology (KO) terms [26]. KO terms were then assigned to KEGG pathways and KEGG BRITE orthologous protein families by mapping these terms to the KEGG Orthology Based Annotation System (KOBAS) database [27]. Enriched KEGG pathways were identified using a cut-off of P $< 0.01$ (Fisher's exact test). KEGG functional enrichments of differentially expressed proteins were integrated and displayed using the program FuncTree [28]. StringApp (v.1.7.1) [29] in the Cytoscape platform (v.3.9.1) [30] was used to conduct protein-protein interaction networking of proteins differentially expressed among three developmental stages (i.e. Ee, El and Af). Since *S. scabiei* data are not included in STRING database, the homologues of the red spider mite, *Tetranychus urticae*, were used for networking. The full STRING network-type was selected and a confidence (score) cut-off of $> 0.9$ implemented. Multi-algorithm cluster analysis was conducted with clusterMaker2 (v.2.2) [31] using the Markov clustering algorithm in Cytoscape. Only clusters containing more than four proteins were included.

## Results

### Proteomes of egg and adult female stages of *S. scabiei*

We identified and quantified a total of 1,761 proteins to represent eggs at an early stage (Ee), eggs at a late stage (El) of embryonation/development and the adult female stage (Af) of *S. scabiei*. This number represents 19.2% of the proteins predicted for *S. scabiei* [16]. Of the 268 (15.2%) orphan proteins (with unknown identity and function) identified, 84.8% (n = 1,493) had relatively conserved homologs in various mites (including *Dermatophagoides pteronyssinus*, *Psoroptes ovis* and *Euroglyphus maynei*) and the flies *Drosophila melanogaster* and *Rhagoletis zephyria* (tephritid).

Most proteins identified were in the El stage (*n* = 1,493), followed by Af (1,386) and Ee (1,261). The full list of proteins identified in individual developmental stages is given in S1 Table, and the numbers unique to, or shared by, stages are shown in a Venn diagram (Fig 1A). Overall, most proteins (*n* = 1382; 78.5%) were shared by two of the three *S. scabiei* stages studied. Of these shared proteins, more than half (*n* = 997; 56.6%) were detected in all three stages. The relative ratios of proteins expressed within individual stages ranged from 2.5% to 15.5%. Notably, the largest stage-specific set of proteins identified was in Af (*n* = 215; 15.5%), followed by El (*n* = 132; 8.8%) and Ee (*n* = 32; 2.5%).

### Comparison of molecular functions

The distribution of the molecular functions (GO level 2) of proteins identified in each developmental stage, according to GO, is summarised in Fig 1B. Most proteins were associated with binding (GO: 0005488; 42.1–46.4%) and catalytic activity (GO: 0003824; 34.8–39.8%) in individual stages. Each of these two major functional categories (binding and catalytic activity) contained at least 480 annotated proteins, whereas proteins involved in structure, transporter and antioxidant activities were less represented. A more detailed appraisal of molecular functions (GO level 3) revealed proteins involved predominantly in the binding of compounds, ions, proteins and small molecules, while hydrolase molecules were represented mainly by the 'catalytic activity' category (S2 Table). In terms of the percentages of proteins classified in each GO sub-category, there was no obvious difference among the three developmental stages studied (Fig 1B).

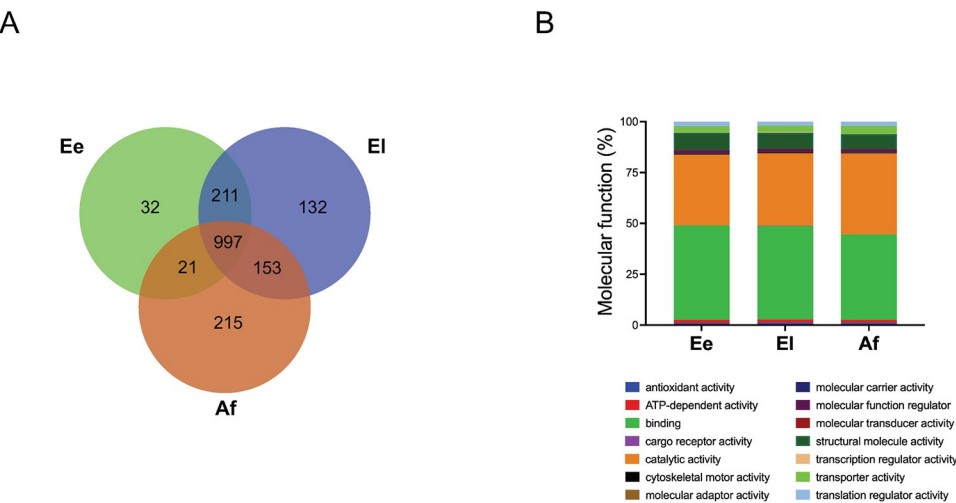

**Fig 1. *Sarcoptes scabiei* proteins and molecular functions.** (**A**) Venn diagram showing the numbers of proteins unique to, or shared by, different developmental stages of *Sarcoptes scabiei*: eggs at an early stage (Ee); eggs at a late stage (El) of embryonation/development; adult female (Af) stage. (**B**) The distribution of the molecular functions (Gene Ontology (GO)–level 2) of proteins quantified in each developmental stage. Distribution was expressed as a percentage of the total number of proteins identified in a particular developmental stage, to allow quantitative comparisons between or among the three stages of *S. scabiei* (see S2 Table).

## Differential protein expression between/among stages

Principal component analysis (PCA) of the proteome of the three developmental stages (Ee, El and Af) of *S. scabiei* showed that individual stages clustered relatively tightly together (Fig 2A). Interestingly, the difference between stages Ee and El was substantial. Following the PCA analysis, hierarchical clustering showed a clear division of the proteomic data set into three distinct groups representing individual stages of *S. scabiei* (Fig 2B). Pairwise comparisons showed extensive differential protein expression among these stages (Fig 3). Markedly more proteins ($n$ = 295) were upregulated in Ee compared with El, and 53 proteins were down-regulated in Ee compared with El (Fig 3A). More differences were observed between Ee and Af than between El and Af. Specifically, 409 proteins were up-regulated and 275 down-regulated in Ee compared with Af (Fig 3B), and 267 were up-regulated and 373 proteins down-regulated in El compared with Af (Fig 3C). The full list of proteins differentially expressed among the three stages of *S. scabiei* (upon pairwise comparison) is given in S3 Table.

## Involvement of differentially expressed proteins in biological pathways and processes

Kyoto Encyclopedia of Genes and Genomes (KEGG) pathway enrichment analysis showed that proteins expressed differentially between at least two developmental stages were involved in one to four biological categories (i.e., cellular processes, genetic information processing, organismal systems and/or metabolism), and linked to key components of parasite growth and metabolism (S4 Table). In the Ee stage, differentially expressed proteins were mainly associated with genetic information processing ($n$ = 24), and a small number with cellular processes ($n$ = 4) and/or metabolism ($n$ = 2) (Fig 4; S4 Table). In the genetic information processing category, transcription (16 proteins in the mRNA surveillance pathway) and DNA replication/ repair (8 proteins) were the two most enriched processes in the Ee stage; four proteins were inferred to link to cell growth or death (cellular processes), and two to nucleotide metabolism (metabolism). In contrast to the Ee stage, none of the differentially expressed proteins was

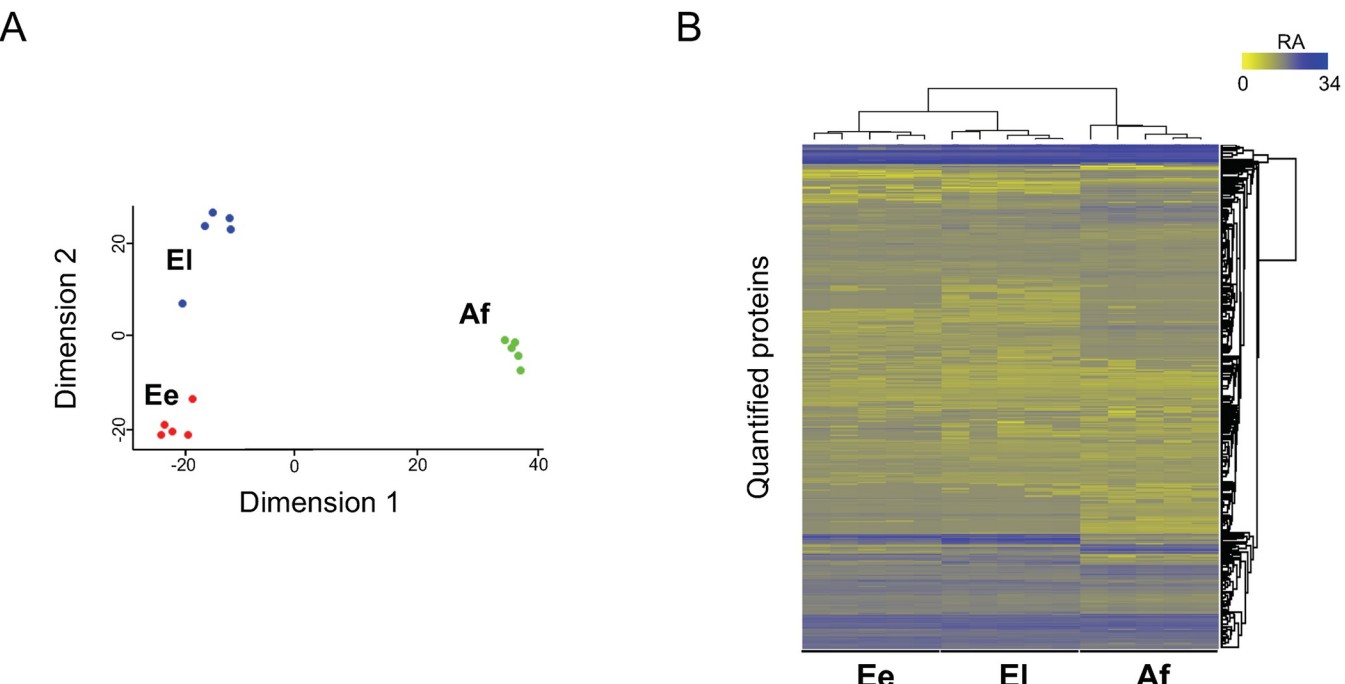

**Fig 2. Analyses of the somatic proteome of *Sarcoptes scabiei*.** (**A**) Principal component analysis (PCA) of the somatic proteomes of eggs at an early stage (Ee), eggs at a late stage (El) of embryonation/development, and adult females (Af) of *Sarcoptes scabiei*, respectively. (**B**) Heatmap displaying the expression profiles for these three distinct developmental stages. Normalised protein abundance is shown in a grey-to-blue scale (i.e., low to high abundance).

involved in cellular processes. Notably, more than 50% (i.e., 27 of 50) of the differentially expressed proteins in El were enriched in metabolic pathways, including energy metabolism ($n = 10$; oxidative phosphorylation) as well as nucleotide ($n = 9$; purine), amino acid ($n = 4$; 4.1 tyrosine) and carbohydrate ($n = 4$; propanoate) metabolism. Similar to the El stage, most

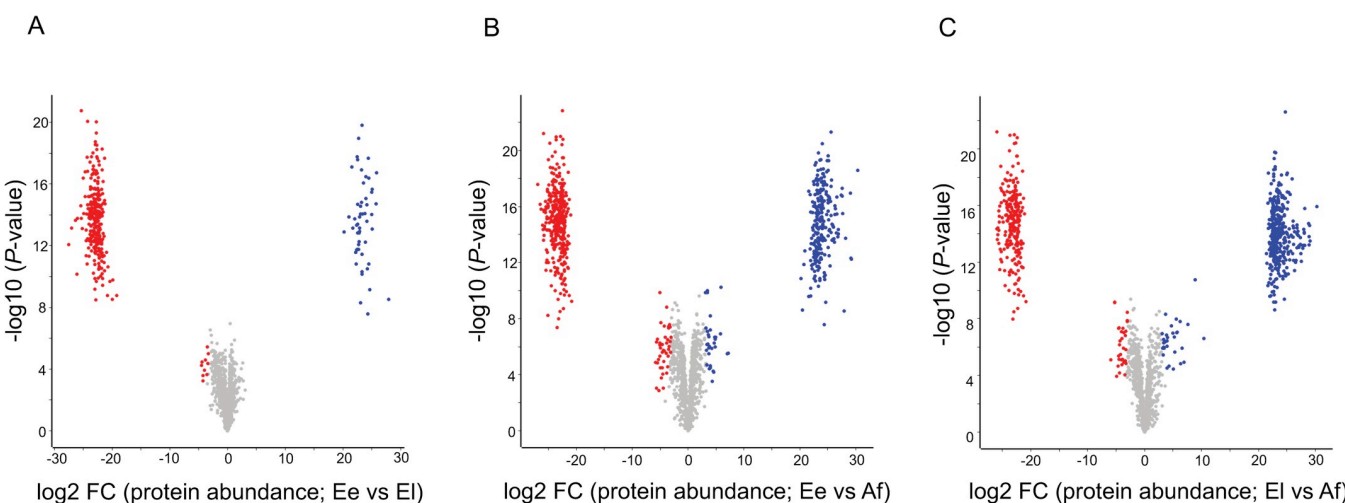

**Fig 3. Volcano plots of proteins differentially expressed among eggs at an early stage (Ee), eggs at a late stage (El) of embryonation/development, and adult females (Af) of *Sarcoptes scabiei* (upon pairwise comparison).** (**A**) Proteins differentially expressed between Ee (up-regulated, red) and El (up-regulated, blue). (**B**) Proteins differentially expressed between Ee (up-regulated, red) and Af (up-regulated, blue). (**C**) Proteins differentially expressed between El (up-regulated, red) and Af (up-regulated, blue).

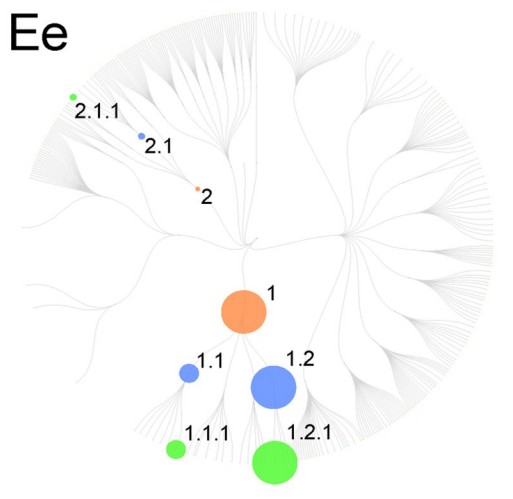

Ee

**Biological category**
**Biological process**
**KEGG pathway**

1. Genetic Information Processing
 1.1 Replication and repair
 1.1.1 DNA replication
 1.2 Transcription
 1.2.1 mRNA surveillance pathway
2. Cellular Processes
 2.1 Cell growth and death
 2.1.1 Cell cycle

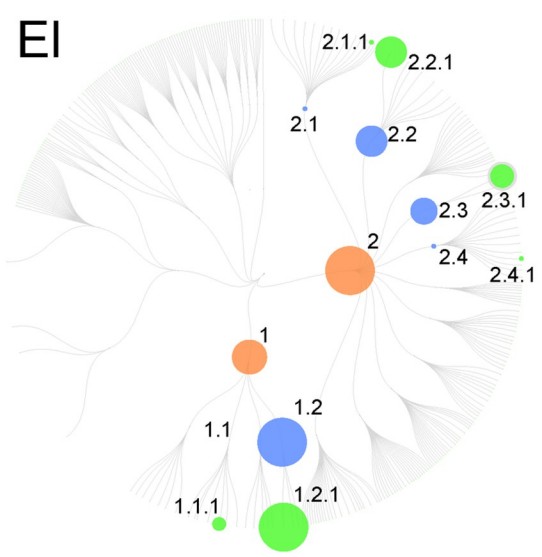

El

1. Genetic Information Processing
 1.1 Folding sorting and degradation
 1.1.1 Protein export
 1.2 Transcription
 1.2.1 Spliceosome
2. Metabolism
 2.1 Carbohydrate metabolism
 2.1.1 Propanoate metabolism
 2.2 Energy metabolism
 2.2.1 Oxidative phosphorylation
 2.3 Nucleotide metabolism
 2.3.1 Purine metabolism
 2.4 Amino acid metabolism
 2.4.1 Tyrosine metabolism

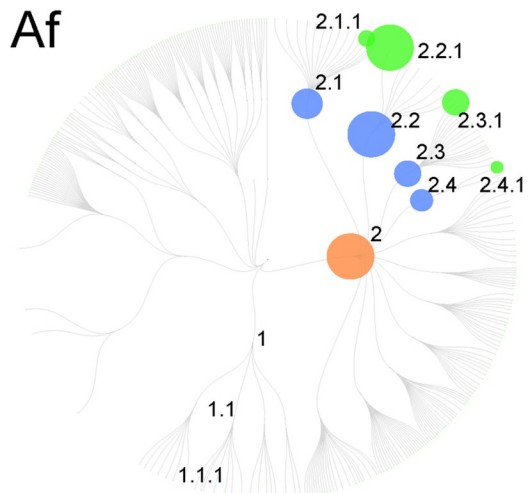

Af

1. Genetic Information Processing
 1.1 Folding sorting and degradation
 1.1.1 Protein export
2. Metabolism
 2.1 Carbohydrate metabolism
 2.1.1 Propanoate metabolism
 2.2 Energy metabolism
 2.2.1 Oxidative phosphorylation
 2.3 Lipid metabolism
 2.3.1 Fatty acid biosynthesis
 2.4 Nucleotide metabolism
 2.4.1 Purine metabolism

**Fig 4. Enriched biological processes and associated pathways (KEGG) of proteins differentially expressed among eggs at an early stage of embryonation/development (Ee), at a late stage (El) of development, and adult females (Af) of *Sarcoptes scabiei* (upon pairwise comparison).** Enriched annotations of highly expressed proteins [fold change (FC) of > 2; false discovery rate (FDR) of < 0.05] are listed. Dot sizes indicate the counts of proteins which are significantly highly expressed (see also S2 Table).

differentially expressed proteins (*n* = 87 of 94; 92.6%) enriched in the Af stage were in the metabolism category (Fig 4), being represented predominantly by energy (*n* = 28), carbohydrate (*n* = 20) and lipid (*n* = 18) metabolism. Least differentially expressed proteins (*n* = 7) enriched in the Af stage were assigned to genetic information processing, and inferred to be involved in protein export (S4 Table).

## Protein-protein interaction networks for differentially-expressed proteins

Protein-protein interaction network analysis inferred distinct interaction modules among the distinct developmental stages (Ee, El and Af) of *S. scabiei* studied here (S5 Table). In total, 246, 424 and 346 differentially expressed proteins with *T. urticae* homologous (S5 Table) were identified in Ee, El and Af, which were used to construct interaction networks. Analyses revealed 5, 9 and 8 protein-protein interaction networks for the Ee, El and Af stages, respectively. For Ee, there were 34 nodes/proteins which formed 5 networks (S5 Table); three networks were associated with genetic information processing, including DNA replication (network 1 with 9 nodes/proteins), RNA processing and splicing (network 1 with 8 nodes/proteins), and nuclear pore complex (network 5 with 5 nodes/proteins). For El, 9 networks were constructed from 62 nodes/proteins (S5 Table); in addition to RNA processing and splicing (network 1 with 14 nodes/proteins), DNA replication (network 2 with 7 nodes/proteins) and nuclear pore complex (network 5 with 5 nodes/proteins), additional networks linked to genetic information processing were identified in this stage, including ribosomal protein (network 4 with 7 nodes/proteins), the SWI/SNF complex (network 5 with 6 nodes/proteins) and DNA-directed RNA polymerase complexes (network 6 with 6 nodes/proteins). Unlike the results for the two egg stages (Ee and El), networks for the Af stage represented predominantly by metabolic functions or energy production, such as mitochondrial protein (network 1), metabolic enzymes (networks 3 and 4), NADH dehydrogenase (network 6) and ATPase complexes (network 7), whereas networks associated with signal recognition (network 2) and muscle movement and structure (network 5) were less conspicuous (S5 Table). Notably, the largest network of mitochondrial complex-related proteins was inferred to contain 26 nodes/proteins.

## Discussion

By defining a proteome for three key developmental stages (Ee, El and Af) of *S. scabiei*, we have been able to provide the first insights, at the protein level, into key biological processes and pathways enriched in stages of this parasitic mite, some of which likely play relevant roles in parasitism on human and other animal hosts. *S. scabiei*, eggs develop *in vitro* from Ee (within ~ 24 h) to El, which hatch within further ~30-48 h to larvae. Larval and nymphal stages establish on host skin, feed on dermal and epidermal tissues as well as secretions, and rapidly grow, develop into adult male or female mites and reproduce within a period of ~10–15 days [32]. During this transition, one might expect profiles of expression for particular protein groups relating to feeding, the acquisition and digestion of essential nutrients, metabolism and body structure and integrity, to meet major demands for mite growth and development on the host animal, and molecular processes (including parasite-host cross-talk) that ensure the survival and reproduction of the mite. These profiles were reflected in the enriched pathways and

protein-protein interaction networks for the proteomes of distinct developmental stages of *S. scabiei*.

The results of the proteomic analysis show substantial variation among the three stages investigated (Fig 1; S1 and S5 Tables). The findings suggest that the adults rely heavily on the catabolism of carbohydrates, lipids, amino acids and nucleotides for survival once the mite is established on the host. Peptidases, including aminopeptidase, carboxypeptidase, endopeptidase, metallopeptidases and serine peptidases, are recognised as crucial molecules in the degradation of skin tissues (epidermis and dermis), sebum and skin exudate, and are likely to be essential for the growth, development and survival of *S. scabiei*. Select aminopeptidases (*n* = 3), dipeptidases (*n* = 3), a metallopeptidase, a serine peptidase, a cysteine peptidase and an aspartic peptidase were conspicuous in one or more of the three stages of *S. scabiei* (S1 Table), in accord with previous transcriptomic findings [16]. We propose that Af-enriched proteins linked to genetic information processing (protein export) are involved in reproductive processes including egg production in *S. scabiei*.

Both the pathway enrichment and protein-protein interaction network analyses revealed a substantial increase in distinct metabolic processes (from 7% to >90%), reflecting major energy and nutrient requirements for growth and development during the transition to the reproductive stage of *S. scabiei*. Enriched pathways included carbohydrate, amino acid, energy and lipid/fatty acid metabolism (Fig 4). Lipid metabolism is up-regulated in *S. scabiei* when this mite reaches the Af stage. While a significant enrichment in lipid-related metabolism was detected in this stage, we identified 18 relatively abundantly expressed proteins to be involved in the fatty acid biosynthesis pathway in Af. Given the intimate association between the mite and skin tissues, we propose that *S. scabiei* upregulates its lipid metabolism (e.g., energy storage lipids—triradylglycerols), because of the availability of essential fatty acids from secretions/excretions and tissues of host skin. This hypothesis warrants future investigation.

*S. scabiei* develops from an egg to a reproductively active adult within ~ 2 weeks, and is associated with rapid embryonic development, metamorphosis and moulting to the protonymph/tritonymph/adult stages. Although only two vitellogenin-like proteins were identified in the mite proteome, these embryonic development-related proteins were recorded as abundantly expressed in all stages studied (i.e. Ee, El and Af). Due to their high aqueous solubility and distinctiveness in sequence (i.e. 48–49%) from homologues of two house dust mite species (*D. farinae* and *D. pteronyssinus*) [33,34], these vitellogenin-like proteins have been considered as diagnostic markers for scabies [35]–a proposal warranting experimental evaluation. Additionally, it is believed that the moulting process in *S. scabiei* is similar to that of other acarids, involving regeneration (i.e. digestion of the old cuticle and production of a new one) and hardening of the cuticle [36].

In the present study, at least 9 cuticular proteins and two collagens were represented in the Af stage, compared with 15 in the egg stages. Although there are differences in the numbers of predicted collagens, 11 of them are shared among all of these stages (Ee, El and Af; S1 Table). These proteins are likely crucial for the maintenance of body shape and integrity by the mite, and for skin burrowing, and contact and cross-talk with the host. An analysis of relative abundance indicated that these collagens and cuticular proteins are expressed mainly in the El and Af stages (S1 Table). All 15 collagens and cuticular proteins identified have significant sequence similarity (45–85%) to homologs in other mites, including *D. farinae*, *D. pteronyssinus*, *Euroglyphus maynei* (dust mites) and *Tyrophagus putrescentiae* (mould or cheese mite) (S6 Table). The collagens of *S. scabiei* are only distantly related to those of cuticle of *D. melanogaster* (i.e., adult cuticle proteins 1 and 65Aa; accession nos. NP_477115 and NP_477282; ref. [37]), indicating significant differences in cuticle synthesis between *S. scabiei* and the vinegar fly. This appears to be reflected in the changes in number and/or abundance of cuticular

proteins (including chitin, chitinase and papilin) from El (n = 6) to Ee (4) and then Af (1) (cf. S1 Table).

In addition, we identified 13 and 15 muscle proteins, such as myosin and troponin, in the proteome of the El and Af stages, respectively, whereas 10 were found in the Ee stage. The abundance of these molecules was consistent with a previous somatic proteomic study of mixed-developmental stages of *S. scabiei* [38], but we have been able to discern that the increased number and abundance in the Af stage (S1 Table) is likely linked to higher demand of muscle movement in the adult stage on and in the skin, particularly during burrowing in the stratum granulosum.

A panel of allergens (*n* = 31), including Sar s 1 and 3 allergens, was identified in *S. scabiei* somatic proteome, consistent with findings from previous molecular studies [16,39,40]. Conspicuous were two protease families, i.e. the scabies mite inactivated serine protease paralogues (SMIPP-S) and scabies mite inactivated cysteine protease paralogues (SMIPP-C), in the different developmental stages (S1 Table). In particular, two SMIPP-S and three SMIPP-C proteins were highly and constitutively expressed (S1 Table), indicating that they play key roles in mite biology and/or parasitism. Published evidence indicates that the SMIPP-S proteins inhibit complement activation and a serine proteases closely related to the SMIPP-S family is involved in the digestion of filament aggregating protein (filaggrin) in human skin [39,41], whereas SMIPP-C molecules have been demonstrated experimentally to promote cutaneous blood coagulation and to inhibit plasmin-induced fibrinolysis [42]. Further work is warranted to explore the precise roles of individual SMIPP-S and SMIPP-C members in mite biology, parasite-host cross-talk and parasitism.

A genomic investigation [16] inferred the genes coding for allergens in *S. scabiei*, 85 of which were specifically transcribed in mixed-developmental stages. The original number of allergens predicted (*n* = 85) was substantially higher than the 31 proteins detected here, 21 of which matched those in proteomic data set representing excretory/secretory proteins (*n* = 236) [16], 15 of which have homologs of known allergens in *D. farinae* and *D. pteronyssinus* [33,34]. The discrepancy in numbers between this proteomic and the previous transcriptomic study [16] is very plausible and can be explained by the detectable proteins being present exclusively in the soma of the mite, and/or by an inability of the present, quantitative proteomic approach to detect minute amounts of proteins in a complex suspension of molecules from the mite. Further work is required to explore the composition and structure of gene families encoding the different types of allergens, and their expression and transcription profiles throughput development, and in mite faeces, to better understand which stages/components/excreta induce the most pronounced allergic responses in people. In addition, a detailed investigation of all developmental stages of the mite to establish the precise nature and composition of the allergenome at the protein and transcriptomic levels would be informative.

## Conclusions

Here, we explored the somatic proteome of egg and adult female stages of the scabies mite–one of the most important cutaneous pathogens of humans globally. Originally, in order to achieve optimal quantitation, we considered using a combination of data-independent acquisition (DIA) and data-dependent acquisition (DDA) methods to quantify mite proteins, but such a workflow would have required larger amounts of mite materials (stages) for protein extraction, which, obviously is a major challenge, in practice, for some stages (particularly eggs) of the mite. Using our shotgun LC-MS/MS-based approach, we quantified a total of 1,761 proteins in these developmental stages and observed a marked proteomic differentiation among these select stages, particularly for molecules that are likely to play essential roles in

development, including genetic information processing (including DNA replication/repair, transcription or protein export), a small number of cellular processes and metabolism (energy/ respiratory, nucleotide, amino acid, carbohydrate and/or lipid), and host-parasite cross-talk. Although the identification of abundant proteins, such as peptidases and SMIPPs, suggested roles for them in parasitism, detailed molecular investigations are needed to test this proposal. More work is also needed to assess the potential of select vitellogenin-like proteins as immuno- or molecular-diagnostic markers and *S. scabiei*-specific proteins as potential drug or vaccine targets. Thus, the present proteomic findings clearly provide some new insight into the biology of this fascinating and highly significant parasitic mite, enhance the value of existing genomic and transcriptomic data sets, and might assist the future discovery of new acaricides against this and related mites, and the study of their modes/mechanisms of action. This work shows clearly the advantages of using a high throughput LC-MS/MS-based methodology for proteomic investigations of tiny and socio-economically significant parasitic acarines.

## Supporting information

**S1 Table. The full list of proteins quantified in the *Sarcoptes scabiei* proteome of different developmental stages (i.e., eggs at an early stage (Ee); eggs at a late stage (El) of embryonation/development; adult female (Af)).**
(XLSX)

**S2 Table. The number of quantified proteins inferred to be involved in the molecular function (levels 2 and 3) in each stage of *Sarcoptes scabiei* (eggs at an early stage (Ee); eggs at a late stage (El) of development; adult female (Af)), according to the Gene Ontology (GO) categories.**
(XLSX)

**S3 Table. The full list of differentially expressed proteins in different developmental stages of *Sarcoptes scabiei* (i.e., eggs at an early stage of development (Ee); eggs at a late stage of development (El) of embryonation/development; and adult female (Af)).**
(XLSX)

**S4 Table. Enriched biological categories, biological processes and KEGG pathways of differentially expressed proteins in different developmental stages of *Sarcoptes scabiei* (i.e., eggs at an early stage (Ee); eggs at a late stage (El) of embryonation/development; and adult female (Af)).**
(XLSX)

**S5 Table. STRING networks for upregulated proteins in the *Sarcoptes scabiei* proteome of key different developmental stages (i.e., eggs at an early stage (Ee); eggs at a late stage (El) of embryonation/development; adult female (Af)).**
(XLSM)

**S6 Table. The full list of quantified collagens and cuticular proteins in the proteome of *Sarcoptes scabiei* representing key different developmental stages (i.e., eggs at an early stage (Ee); eggs at a late stage (El) of embryonation/development; and adult female (Af)).**
(XLSX)

## Acknowledgments

The authors thank Sara Taylor for helping with mite isolation, and Chelsea Baker, Sheree Boisen, Scott Cullen and Milou Dekkers at the Queensland Animal Science Precinct, University

of Queensland, Gatton Campus, Australia, for maintaining the porcine *S. scabiei* model, and Yuanting Zheng at the University of Melbourne for some bioinformatic support.

## Author Contributions

**Conceptualization:** Robin B. Gasser, Katja Fischer.

**Data curation:** Tao Wang, Robin B. Gasser, Pasi K. Korhonen.

**Formal analysis:** Tao Wang, Robin B. Gasser, Pasi K. Korhonen.

**Funding acquisition:** Robin B. Gasser, Katja Fischer.

**Investigation:** Tao Wang, Pasi K. Korhonen, Ching-Seng Ang, Guangxu Ma, Gangi R. Samarawickrama, Deepani D. Fernando, Katja Fischer.

**Methodology:** Tao Wang, Deepani D. Fernando.

**Project administration:** Robin B. Gasser, Katja Fischer.

**Resources:** Robin B. Gasser, Katja Fischer.

**Supervision:** Robin B. Gasser, Katja Fischer.

**Validation:** Tao Wang, Robin B. Gasser.

**Visualization:** Tao Wang.

**Writing – original draft:** Tao Wang, Robin B. Gasser.

**Writing – review & editing:** Tao Wang, Robin B. Gasser, Pasi K. Korhonen, Neil D. Young, Ching-Seng Ang, Nicholas A. Williamson, Guangxu Ma, Deepani D. Fernando, Katja Fischer.

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
