## [Decision Letter · Decision Letter 0]

9 Aug 2022

Dear Dr Wang,

Thank you very much for submitting your manuscript "Proteins differentially expressed in Sarcoptes scabiei between eggs and female adults are predominantly involved in genetic information processing and metabolism" for consideration at PLOS Neglected Tropical Diseases. As with all papers reviewed by the journal, your manuscript was reviewed by members of the editorial board and by several independent reviewers. In light of the reviews (below this email), we would like to invite the resubmission of a significantly-revised version that takes into account the reviewers' comments. 

We cannot make any decision about publication until we have seen the revised manuscript and your response to the reviewers' comments. Your revised manuscript is also likely to be sent to reviewers for further evaluation.

Sincerely,

Jo Middleton

Guest Editor

Aysegul Taylan Ozkan

Section Editor

Reviewer's Responses to Questions

**Key Review Criteria Required for Acceptance?**

**Methods**

-Are the objectives of the study clearly articulated with a clear testable hypothesis stated?

-Is the study design appropriate to address the stated objectives?

-Is the population clearly described and appropriate for the hypothesis being tested?

-Is the sample size sufficient to ensure adequate power to address the hypothesis being tested?

-Were correct statistical analysis used to support conclusions?

-Are there concerns about ethical or regulatory requirements being met?

Reviewer #1: 1. Study methods are all very solid and well described in the text. Sample sizes also seem sufficient and statistical analysis is very robust.

2. Can the authors confirm if the adult females selected for the analysis also contained eggs or if care was taken to select non-gravid Af mites in order to prevent cross contamination of egg proteins into the Af proteome? 

3. Raw data has been deposited in PRIDE but no accession is provided so it is not possible to check this at this stage, a full accession should be provided ahead of publication.

Reviewer #2: The study methodology is well mentioned. The objective of the study is well described and hypothesis is novel. Although the sample size calculation is not mentioned, the nature of the study and the number of the proteins analyzed appears adequate. The statistical analysis used is appropriate.

**Results**

-Does the analysis presented match the analysis plan?

-Are the results clearly and completely presented?

-Are the figures (Tables, Images) of sufficient quality for clarity?

Reviewer #1: Results are clearly described but please see General Comments section for a more detail assessment of what is required to add further impact to the manuscript. Figures are good and clearly presented but Figure 2B could use a more obvious colour scheme in order to more clearly show the variation in gene expression between life cycle stages and replicates - perhaps a blue/yellow scheme would be more obvious.

Reviewer #2: The analysis results presented are in the line of the study objective

**Conclusions**

-Are the conclusions supported by the data presented?

-Are the limitations of analysis clearly described?

-Do the authors discuss how these data can be helpful to advance our understanding of the topic under study?

-Is public health relevance addressed?

Reviewer #1: The conclusions are clearly supported by the results of the study, however I really do feel that more can be made of the analysis by adding further discussion of the individual proteins identified and also through an interaction network to really draw out the pathways and protein connections involved between stages. See General Comments section for further details

Reviewer #2: The conclusion presented is more like a summary. It can be modified to include their main findings and its public health implications along with limitation and future direction of research.

**Editorial and Data Presentation Modifications?**

Reviewer #1: A few minor editorial changes are required as described here:

1. Line 37: "Scabies is a neglected tropical disease......"

2. Line 56: "Scabies is a disease caused by the ectoparasitic mite Sarcoptes scabiei...."

3. Line 93: Remove: "one of the most important parasitic mites of humans and other animals (e.g. pig, dog and wombat) worldwide"

4. Line 98: "defining the proteome of the key developmental stages...."

5. Line 112: I'm not sure that production is the correct terminology for "rearing/maintaining" scabies mites.

6. Line 124: The title of this section is repeated from the previous section and needs to be changed to "Protein extraction and characterisation" or something similar..

7. Line 344: ".. in D. farinae and ...."

Reviewer #2: None

**Summary and General Comments**

Reviewer #1: This is a very well written and well described article providing a really excellent characterisation of the Scabies mite proteome across key reproductive life cycle stages. The study has been conducted very well and the analysis and results presented are robust and clearly add significantly to the existing scientific knowledge of this highly important mite. 

However, in my opinion the paper does require some further analysis/exploitation of the existing data before it can be considered suitable for publication in PLOS NTD. In particular more needs to made of the really impressive proteomic dataset that was generated from the study and this should involve at a minimum a protein-protein interaction network, for example as can be achieved with a tool such as StringDB. Whilst I appreciate that Sarcoptes scabiei is not annotated within StringDB then the annotation for a closely related mite such Tetranychus urticae should be used instead. I feel that this analysis will be important to provide further information on the connections between the proteins identified, especially in the context of the differential expression analysis between life cycle stages. 

This also comes on to my second point, which is the current lack of in depth discussion around specific protein classes and families that were identified from the differential expression analysis as well as from the standard characterisation of the proteome of the mite. For example, there is little if any discussion around key allergens, i.e. house dust mite homologues, vitellogenins and other proteins involved in reproduction and nutrient provision for the developing embryos and also cuticular proteins and structural muscle proteins, i.e. myosin/troponins etc. Also further discussion is required around the enzyme classes identified, including the scabies mote pseduoproteases, SMIPPs which were identified in the initial analysis. The current manuscript title and the analysis of results presented are underwhelming and I feel that a lot more can be made of this impressive dataset by carrying out the further analysis described above.

Finally, Figure 2 may be further improved by adding hierarchical clustering of proteins at both the sample (column) and protein (row) levels as this may also help to identify clusters of proteins between life cycle stages.

The conclusions section then needs to be made stronger, for example what outcomes from the study will help to identify novel acaricides in the future?

Reviewer #2: The authors have analyzed proteins form S. scabiei to find the relation of such proteins with different developmental stages. The liquid chromatography–mass spectrometry technology was used for detection of such proteins is using. The study methodology is well mentioned and results are well presented. However, the abstract presented can be structured and and limitation of the study can be mentioned.

PLOS authors have the option to publish the peer review history of their article (what does this mean?). If published, this will include your full peer review and any attached files.

Reviewer #1: Yes: Stewart Burgess

Reviewer #2: No
---

## [Decision Letter · Decision Letter 1]

10 Nov 2022

Dear Dr Wang,

Thank you very much for submitting your manuscript "Proteomic analysis of Sarcoptes scabiei reveals that proteins differentially expressed between eggs and female adult stages are involved predominantly in genetic information processing, metabolism and/or host-parasite interactions" for consideration at PLOS Neglected Tropical Diseases. As with all papers reviewed by the journal, your manuscript was reviewed by members of the editorial board and by several independent reviewers. The reviewers appreciated the attention to an important topic. Based on the reviews, we are likely to accept this manuscript for publication, providing that you modify the manuscript according to the review recommendations. 

Sincerely,

Aysegul Taylan Ozkan, M.D., Ph.D.,

Section Editor

Aysegul Taylan Ozkan

Section Editor

Reviewer's Responses to Questions

**Key Review Criteria Required for Acceptance?**

**Methods**

-Are the objectives of the study clearly articulated with a clear testable hypothesis stated?

-Is the study design appropriate to address the stated objectives?

-Is the population clearly described and appropriate for the hypothesis being tested?

-Is the sample size sufficient to ensure adequate power to address the hypothesis being tested?

-Were correct statistical analysis used to support conclusions?

-Are there concerns about ethical or regulatory requirements being met?

Reviewer #1: (No Response)

Reviewer #3: Although no clearly articulated single hypothesis is tested in the paper, this work is an example of a well-prepared, novel study presenting new knowledge essential in human disease studies/epidemiology and general mite biology and parasitology. Nonetheless, the work has an ambitious aim, which has been accomplished. Only the presentation of the problem and the new data obtained provide a source for hypotheses that can be tested in further research, as the authors have suggested in the Discussion. I advise that one sentence be included to make it even more explicit that this work can be (and probably will be) a starting point for further, more detailed, or, i.e., purely experimental studies.

The study design is appropriate to address the stated objectives. Of course, more methods and objectives could be stated, but it is impossible to carry out so many tasks in one study (this statement applies to all studies).

There is no information why exactly such a sample size has been analysed but taking into account the prevalence of parasitic mites in general, difficulties in obtaining a decent number of individuals, etc. The sample size seems to be adequate for the conducted study.

At least, as far as my knowledge and the authors' descriptions of methodological approaches provided, together with references to those commonly used and practised in a similar type of research, allow me to conclude I see no objection to the methods used in this investigation. In addition, the authors provided information and results of the relevant statistical analysis.

The authors supplied information on ethics approval due to the study with animals. It fulfils all the ethical requirements regarding the type of studies conducted.

**Results**

-Does the analysis presented match the analysis plan?

-Are the results clearly and completely presented?

-Are the figures (Tables, Images) of sufficient quality for clarity?

Reviewer #1: (No Response)

Reviewer #3: The analysis performed during the research and after that presented in the Results section of the manuscript adequately matches the plan outlined in its previous parts.

It is worth noting the thorough presentation of the results, with many references to the extracted data constituting the supplementary material and that deposited in the public database. Also, unexaggerated comparisons and explanations of submitted figures indicate a satisfactory reporting of the exciting findings.

Figures are of sufficient general quality. The data in the tables are precise, mainly when explained directly in the manuscript. However, I must say that:

a) shades of colours in Figure 1B may be challenging to read, especially for people with impaired vision, as some are pretty similar. I would consider changing the colours to obtain better contrast before releasing the potential publication so that the results presented were not questionable,

b) Fig. 2B should be stretched vertically because the branches of the tree are remarkably close together, hence quite unreadable,

c) and the last issue of the graphical representation of biological processes and associated protein pathways in Fig. 4. I use the 4K monitor, and I had a problem seeing them at all at first glimpse. The lines of those pathways should be thicker.

I think that my proposed corrections may improve the readability of graphs for the readers.

**Conclusions**

-Are the conclusions supported by the data presented?

-Are the limitations of analysis clearly described?

-Do the authors discuss how these data can be helpful to advance our understanding of the topic under study?

-Is public health relevance addressed?

Reviewer #1: (No Response)

Reviewer #3: The presented data support conclusions.

There is much to say about the limitations, although the authors provided some degree of it. The sentence “Originally…” (line 428) would be better if provided as an example of such limitation instead of unsuccessful original plans (maybe you will enrich the topic soon).

The authors discuss how these data can help advance our understanding of the topic under study.

The public health relevance is mentioned, but it only makes an impression as a formal addition. Meanwhile, the research presented in this manuscript shows how many exciting things remain to be discovered about the parasitic mite of man, which was probably first mentioned in the literature by Aristotle. I suggest adding something more related to public health here.

**Editorial and Data Presentation Modifications?**

Reviewer #1: (No Response)

Reviewer #3: Lines 68-72. Some reference would be welcomed.

Line 87. I think there should be “mite genomic….”

Line 117. Production might be the correct word for, e.g., pigs if they are sold for meat. Also, even mass production of predatory mites of the family Phytoseiidae is used for natural control of mainly spider mites (Tetranychidae) as they damage many cultivated plants. In this case (even if there are examples in some literature), “production” is an incorrect term, and I advise the word “rearing” instead of the former one.

Line 360. Sarcoptes is the genus name. Also, we do not know if the results can be expanded to other species classified to it. Hence, “S. scabiei individuals/representatives/…” sounds better without ambiguities.

Line 395. For the general reader, it would be helpful to explain what “stratum granulosum” is. The same is true for “cross-talk” (lines 337, 378, 408, 438).

Lines 93, 369, 413. There are mistakes in the specific epithet; please replace “farina” with “farinae.”

**Summary and General Comments**

Reviewer #1: The authors have clearly addressed the comments raised in the review and I would now be happy to see this paper published in its current form.

Reviewer #3: The manuscript submitted, in addition to the enormous effort to deliver the research results undertaken therein, is worthy of being considered for publication in the PLOS Neglected Tropical Diseases journal. The mistakes are only nominal, most of them in the form of editorial amendments. However, the copyeditors could not cope with this task given the poorly represented scientific field. Nevertheless, the oversights found, or details that need ONLY MINOR improvements, oblige me to propose a 'Minor review' as my recommendation for the Editor at that moment. My comments and suggestions are intended to ensure that the contributors will publish a paper of the highest possible quality. After correcting and considering a few things already mentioned in my review, the manuscript should be accepted. 

I agree to disclose my name to the authors to build better, transparent science that is a core for the PLOS non-profit corporation. I declare that I have no conflict of interest. I congratulate the authors for their excellent work and the Editor for inviting me to review.

PLOS authors have the option to publish the peer review history of their article (what does this mean?). If published, this will include your full peer review and any attached files.

Reviewer #1: No

Reviewer #3: Yes: Mateusz Zmudzinski

Figure Files:

Data Requirements:

Reproducibility:

References

---

## [Editor Report · Decision Letter 2]

14 Nov 2022

Dear Dr Wang,

We are pleased to inform you that your manuscript 'Proteomic analysis of Sarcoptes scabiei reveals that proteins differentially expressed between eggs and female adult stages are involved predominantly in genetic information processing, metabolism and/or host-parasite interactions' has been provisionally accepted for publication in PLOS Neglected Tropical Diseases.

Best regards,

Aysegul Taylan Ozkan, M.D., Ph.D.,

Section Editor

Aysegul Taylan Ozkan

Section Editor

---

## [Editor Report · Acceptance letter]

18 Nov 2022

Dear Dr Wang,

We are delighted to inform you that your manuscript, "Proteomic analysis of Sarcoptes scabiei reveals that proteins differentially expressed between eggs and female adult stages are involved predominantly in genetic information processing, metabolism and/or host-parasite interactions," has been formally accepted for publication in PLOS Neglected Tropical Diseases.

Best regards,

Shaden Kamhawi

co-Editor-in-Chief

Paul Brindley

co-Editor-in-Chief
